# An isoform-specific function of Cdc42 in regulating mammalian Exo70 during axon formation

Priyadarshini Ravindran[1] , Andreas W Püschel[1,2]

**The highly conserved GTPase Cdc42 is an essential regulator of cell polarity and promotes exocytosis through the exocyst complex in budding yeast and *Drosophila*. In mammals, this function is performed by the closely related GTPase TC10, whereas mammalian Cdc42 does not interact with the exocyst. Axon formation is facilitated by the exocyst complex that tethers vesicles before their fusion to expand the plasma membrane. This function depends on the recruitment of the Exo70 subunit to the plasma membrane. Alternative splicing generates two Cdc42 isoforms that differ in their C-terminal 10 amino acids. Our results identify an isoform-specific function of Cdc42 in neurons. We show that the brain-specific Cdc42b isoform, in contrast to the ubiquitous isoform Cdc42u, can interact with Exo70. Inactivation of Arhgef7 or Cdc42b interferes with the exocytosis of post-Golgi vesicles in the growth cone. Cdc42b regulates exocytosis and axon formation downstream of its activator Arhgef7. Thus, the function of Cdc42 in regulating exocytosis is conserved in mammals but specific to one isoform.**

## Introduction

The highly conserved GTPase Cdc42 has a well-established function in regulating actin dynamics and the polarization of cells (Garvalov et al, 2007; Chen et al, 2012). Alternative splicing generates two Cdc42 isoforms in vertebrates that differ only in their C-terminal 10 amino acids, resulting in differences in their lipid modification. The ubiquitously expressed Cdc42 isoform (Cdc42u) is prenylated at the cysteine residue in the CaaX motif, whereas the brain-specific isoform (Cdc42b) contains two C-terminal cysteines that can be palmitoylated (Kang et al, 2008; Nishimura & Linder, 2013). Both isoforms are detectable in the axon of differentiating neurons (Wirth et al, 2013; Mukai et al, 2015; Lee et al, 2021). In mature neurons, Cdc42b is required for the formation of dendritic spines (Kang et al, 2008; Wirth et al, 2013, 2022; Yap et al, 2016). It is poorly understood how these isoforms perform their different functions and whether these depend on their interaction with specific effectors.

Cdc42 is a central regulator of neuronal polarity that promotes axon formation by increasing actin turnover (Garvalov et al, 2007). A knockout of Cdc42 that eliminates both isoforms prevents axon formation, and both Cdc42u and Cdc42b are required for normal neuronal polarity (Garvalov et al, 2007; Mukai et al, 2015; Yap et al, 2016). We have previously shown that the Cdc42 GEF Arhgef7 is essential for axon formation (López Tobón et al, 2018). Arhgef7 activates TC10 in a heterologous system indicating that it acts as a GEF for TC10. TC10 regulates membrane expansion by recruiting the exocyst subunit Exo70 to the plasma membrane, where it tethers specialized plasmalemma precursor vesicles (PPVs) to promote axon formation (Dupraz et al, 2009; Gracias et al, 2014). The exocytosis of these PPVs is required for membrane expansion to allow the extension of axons, which requires TC10, Exo70, and other subunits of the exocyst complex (Dupraz et al, 2009; Lalli, 2009; Bustos Plonka et al, 2022). The octameric exocyst complex has a conserved function in polarized exocytosis, and homologs are present from yeast and plants to humans (Koumandou et al, 2007; Martin-Urdiroz et al, 2016). In yeast, the exocyst is recruited to the membrane by Cdc42p, whereas mammalian Cdc42 was shown not to interact with Exo70 (Inoue et al, 2003; Wu et al, 2010). Instead, TC10 recruits the exocyst to the plasma membrane in mammals (Inoue et al, 2003). The inability of Cdc42 to interact with Exo70 may indicate that this function switched to TC10 with its emergence in chordates (Boureux et al, 2007). The inability of Cdc42 to interact with Exo70 was surprising because it shares many effectors with TC10 (Neudauer et al, 1998; Joberty et al, 1999). Here, we show that the palmitoylated Cdc42b isoform is able to interact with Exo70. Cdc42b is required for exocytosis and promotes axon formation downstream of Arhgef7. Thus, the function of Cd42 in regulating Exo70 is also conserved in vertebrates but restricted to one of its isoforms.

## Results and Discussion

### Arhgef7 regulates membrane expansion

We have shown previously that Arhgef7 functions upstream of TC10 during neuronal polarization (López Tobón et al, 2018). Cultures of cortical neurons from the embryonic brain of a cortex-specific

---

[1]Institut für Integrative Zellbiologie und Physiologie, Westfälische Wilhelms-Universität, Münster, Germany [2]Cells-in-Motion Interfaculty Center, University of Münster, Münster, Germany

Correspondence: apuschel@uni-muenster.de

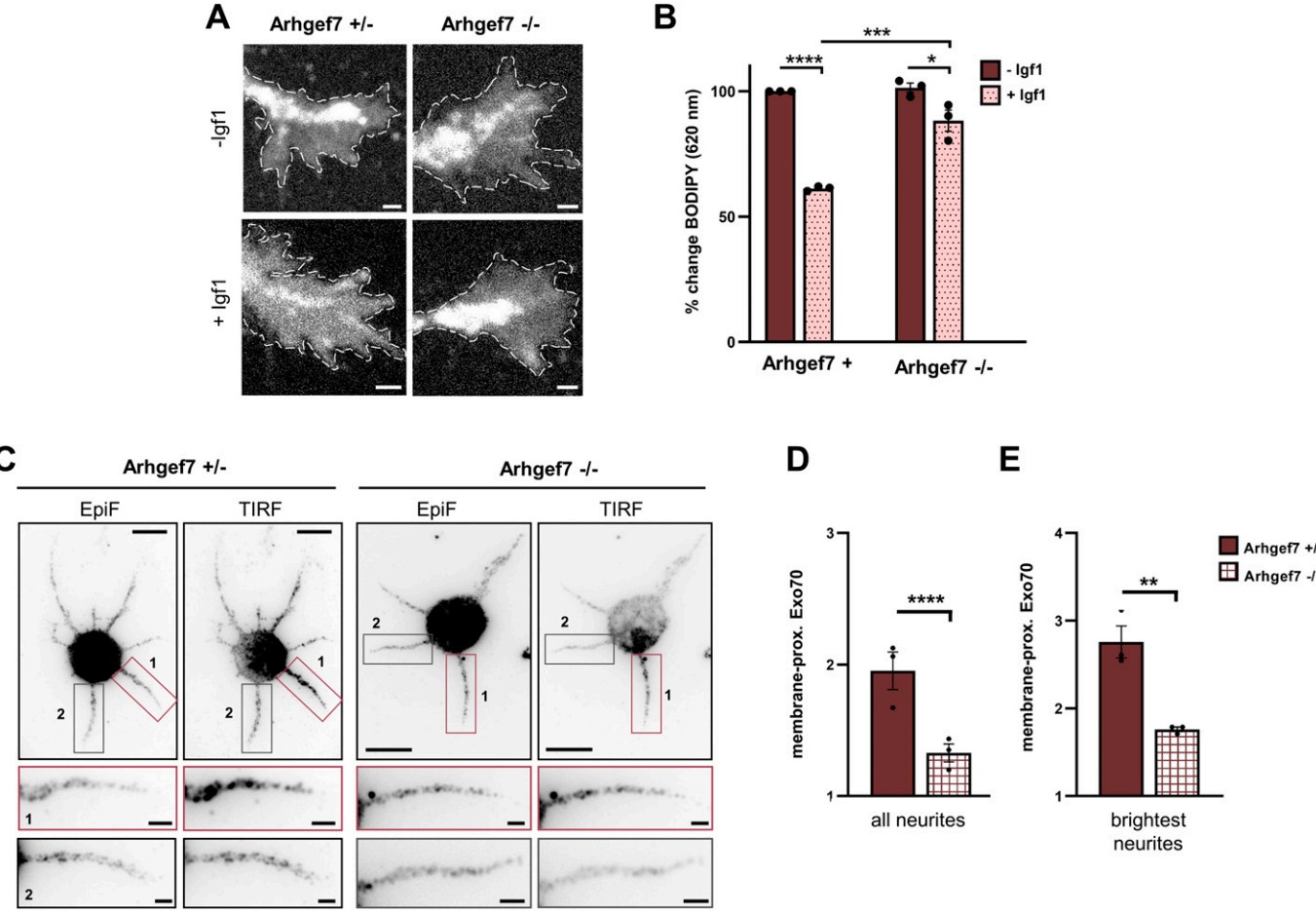

**Figure 1. Loss of Arhgef7 impairs the exocytosis of post-Golgi vesicles.**
**(A)** Cortical neurons from E17 *Arhgef7*^flox/+;*Emx1*^Cre/+ (Arhgef7+/−) and *Arhgef7*^flox/flox;*Emx1*^Cre/+ (Arhgef7−/−) embryos were incubated with BODIPY FL C5-ceramide at 1 d.i.v. to label post-Golgi vesicles and cultured in medium without (−Igf1) or with (+Igf1) 20 nM Igf1 for 10 min. A decrease in the fluorescence intensity at 620 nm indicates the exocytosis of vesicles in growth cones. Scale bar: 2 μm. **(B)** Igf1-stimulated exocytosis was quantified for (A) as the relative change in the fluorescence intensity for BODIPY-labeled vesicles in growth cones at 620 nm, normalized to the signal at 515 nm. The value for control neurons (−Igf1) was set as 100% (one-way ANOVA followed by Tukey's test, N = 45 neurons from each condition from n = 3 independent experiments that are biological replicates). **(C)** Cortical neurons from Arhgef7+/− and Arhgef7−/− E17 embryos were analyzed at 1 d.i.v. by staining with an anti-Exo70 antibody. Epifluorescence and TIRF signals indicate total and plasma membrane-proximal Exo70, respectively. The neurite with the highest Exo70 signal (1) and a second neurite (2) are displayed at a higher magnification. Scale bar: 10 and 2 μm, respectively. **(D, E)** Membrane-proximal Exo70 was quantified for (C) at 1 d.i.v. in cortical neurons from Arhgef7+/− and Arhgef7−/− embryos. The intensity of the total internal reflection fluorescence microscopy signal in neurites was normalized to epifluorescence signals (total Exo70) and area. The average relative intensity for membrane-proximal Exo70 in all neurites (D) and the brightest neurites (E) was quantified (an unpaired *t* test, N = 42 neurons from n = 3 independent experiments that are biological replicates). Values are means ± SEM. ns, *P* > 0.05; **P* ≤ 0.05; ***P* ≤ 0.01; ****P* ≤ 0.001; and *****P* ≤ 0.0001. Source data are available for this figure.

*Arhgef7* knockout show a severe impairment in axon formation. To test whether Arhgef7 regulates the exocytosis of post-Golgi vesicles, we analyzed cortical neurons from homozygous cortex-specific *Arhgef7* E17.5 knockout embryos (Arhgef7-cKO: *Arhgef7*^flox/flox; *Emx1*^Cre/+) and compared those with heterozygous Arhgef7-cKO neurons as control (*Arhgef7*^flox/+;*Emx1*^Cre/+). Post-Golgi vesicles were labeled with BODIPY FL C$_5$-ceramide at 1 day in vitro (d.i.v.) (Wang et al, 2011; Xu et al, 2014). After allowing the uptake of BODIPY-ceramide, exocytosis was stimulated by Igf1 and quantified as the decrease in the fluorescence signal at 620 nm for BODIPY-labeled vesicles in the growth cone (Laurino et al, 2005; Wang et al, 2011). Igf1 stimulated the exocytosis of labeled vesicles in heterozygous Arhgef7 c-KO neurons compared with untreated controls as shown by a decrease of 39% in the BODIPY signal (Fig 1A and

B). Igf1-induced exocytosis was significantly reduced but not completely abolished in neurons from homozygous Arhgef7-cKO embryos compared with heterozygous controls. Thus, Arhgef7 is required for the Igf1-induced exocytosis of post-Golgi vesicles.

To confirm that the loss of Arhgef7 affects Exo70, we analyzed the levels of endogenous Exo70 at the plasma membrane in neurons from Arhgef7-cKO embryos by total internal reflection fluorescence microscopy. Cortical neurons were analyzed at 1 d.i.v. by immunofluorescence with an antibody for Exo70. Membrane-proximal Exo70 was quantified using the TIRF signal and normalized to total Exo70 determined by the epifluorescence signal (Fig 1C). There was a marked reduction in membrane-proximal Exo70 in all neurites of homozygous Arhgef7-cKO neurons compared with heterozygous neurons (Fig 1D). This was also evident when the average was

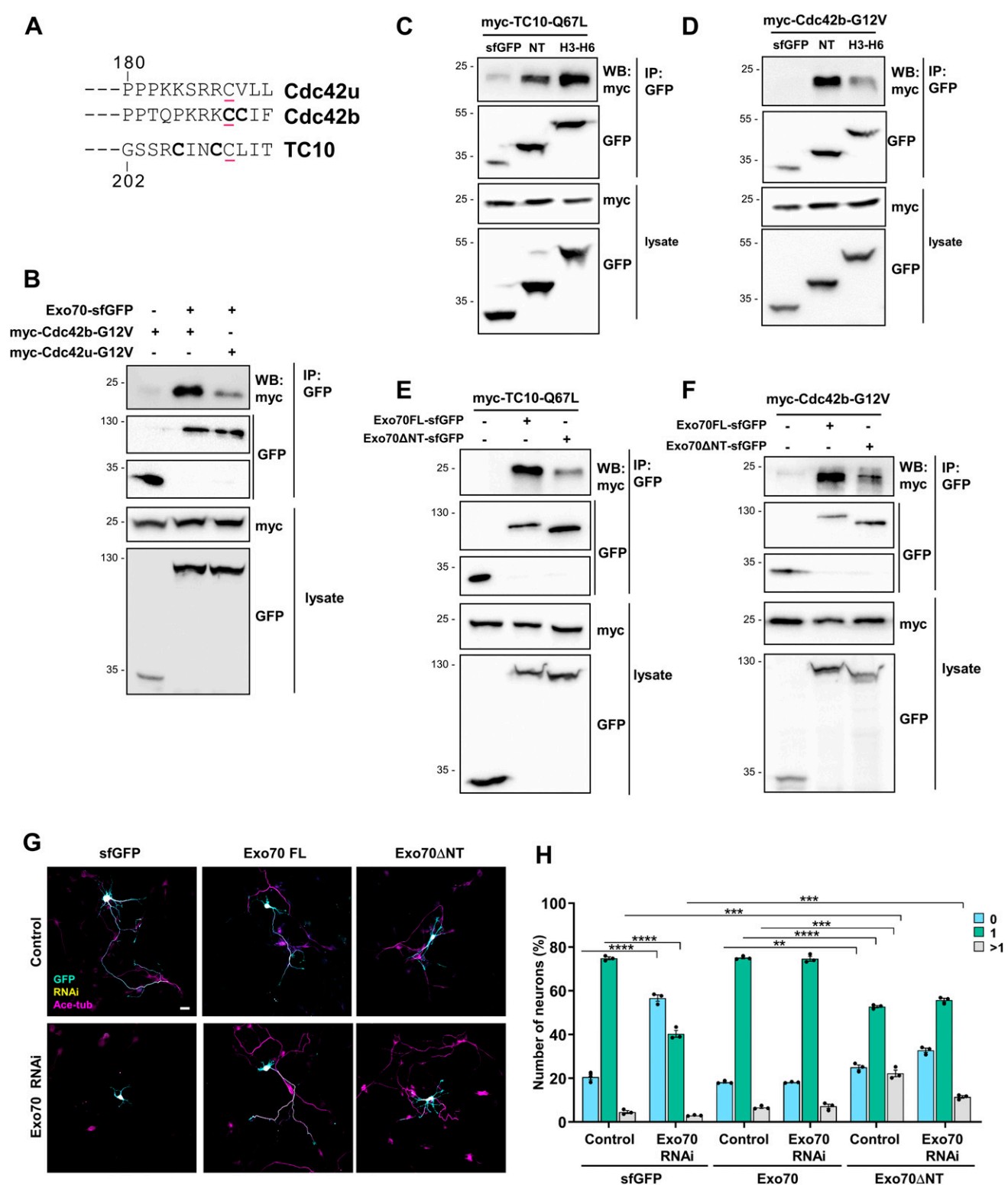

**Figure 2. Cdc42b isoform interacts with Exo70.**
**(A)** Alternative splicing generates the ubiquitous (Cdc42u) and brain-specific (Cdc42b) Cdc42 isoforms, which are prenylated (underlined) and palmitoylated (bold).
**(B)** Exo70 interacts with Cdc42b. sfGFP (control) or Exo70-sfGFP was co-expressed with myc-Cdc42b-G12V or myc-Cdc42u-G12V in HEK293T cells and immunoprecipitated (IP: anti-GFP nanobody), and bound Exo70 was detected by Western blot (WB). The molecular weight is indicated in kD. Blots (IP: GFP) were cropped to show signals for sfGFP and Exo70-sfGFP. **(C, D)** sfGFP-Exo70-NT or sfGFP-Exo70-H3-H6 was co-expressed with myc-TC10-Q67L (C) or myc-Cdc42b-G12V (D) in HEK293T cells and immunoprecipitated, and interacting proteins were detected (WB: anti-myc antibody). **(E, F)** Full-length Exo70-sfGFP or truncated Exo70ΔNT-sfGFP was co-expressed with myc-TC10-Q67L (E) or myc-Cdc42b-G12V (F) in HEK293T cells and immunoprecipitated. Blots (IP: GFP) were cropped to show signals for sfGFP, Exo70-sfGFP, and Exo70ΔNT-

determined only for neurites with the strongest signal for each neuron (Fig 1E). These results indicate that Arhgef7 regulates the plasma membrane localization of Exo70 and the exocytosis of vesicles in developing neurons.

## Isoform-specific interaction between Cdc42 and Exo70

TC10 recruits the exocyst subunit Exo70 to the plasma membrane and was the only GTPase tested that bound to Exo70, whereas Cdc42 did not show an interaction (Inoue et al, 2003; Dupraz et al, 2009). Because only the Cdc42u isoform was tested in these studies, we analyzed the interaction of Exo70 with the Cdc42b isoform, which is palmitoylated like TC10 (Fig 2A). After co-expression with full-length Exo70 in HEK293T cells, Cdc42b could be co-immunoprecipitated with Exo70 (Fig 2B). In contrast, Cdc42u showed only a very weak interaction as reported before (Inoue et al, 2003).

Exo70 is composed of an N-terminal domain predicted to contain a coiled-coil and a series of 19 α-helices that form a long rod (Moore et al, 2007; Mei et al, 2018). The binding site of TC10 was mapped previously to the N-terminal half of Exo70 (Inoue et al, 2003). Guided by its structure, we subdivided Exo70 into several fragments (Fig S1A). The interaction of these fragments with TC10 or Cdc42b was analyzed by co-immunoprecipitation after expression in HEK293T cells to delineate their binding sites (Fig S1B and C). We could identify two Exo70 fragments that interacted but differed in their preference for the GTPases. Both Cdc42b and TC10 showed a strong binding to the N-terminal 84 amino acids (NT) and a fragment spanning helices 3 to 6 (Exo70-H3-H6: amino acids 133–240). These regions are located within the Exo70 fragment that was previously identified as the TC10 binding region (Inoue et al, 2003). TC10 interacted preferentially with Exo70-H3-H6 (Fig 2C), whereas Cdc42b showed a more pronounced binding to Exo70-NT (Fig 2D). The deletion of the N-terminal 84 amino acids (Exo70ΔNT) impaired but did not abolish the interaction of Cdc42b or TC10, confirming that this was indeed a binding site for both of these GTPases (Fig 2E and F). These results identify two domains in Exo70 within this sequence that interact with Cdc42b and TC10.

The exocyst complex tethers PPVs at the plasma membrane before their fusion (Heider & Munson, 2012; Zhu et al, 2017; Ahmed et al, 2018; An et al, 2021). To test the importance of the N-terminal GTPase binding site (Exo70-NT) for the function of Exo70 in neurons, we attempted to rescue the knockdown of Exo70 with Exo70ΔNT. We generated a knockdown construct that targets the exocyst subunit Exo70 and confirmed its efficiency by Western blot after expression in HEK293T cells (Fig S1D and E). Consistent with previous studies, a knockdown of Exo70 in cultured hippocampal neurons severely impairs axon formation, which was restored by the expression of human Exo70-sfGFP, confirming the specificity of the knockdown (Fig S1H and I)

(Dupraz et al, 2009). Hippocampal neurons from E18 rat embryos were transfected at 0 d.i.v. and Golgi-derived vesicles labeled with BODIPY-ceramide at 1 d.i.v. (Fig S1F and G). The loss of Exo70 led to a pronounced impairment of Igf1-induced exocytosis of post-Golgi vesicles consistent with previous results (Dupraz et al, 2009). The expression of full-length Exo70 rescued the suppression of Exo70 and reduced the number of unpolarized neurons from 56.6% ± 1.4% after knockdown to 18.2% ± 0.2%, similar to controls (20.6% ± 1.3% unpolarized neurons). In contrast, Exo70ΔNT rescued axon formation only partially, reducing the number of unpolarized neurons only to 32.7% ± 1% (Fig 2G and H). The expression of Exo70ΔNT alone reduced the number of neurons with a single axon, indicating a weak dominant-negative effect. Taken together, these results reveal an isoform-specific interaction of Cdc42b with Exo70. The first GTPase binding site (Exo70-NT) is important for the function of Exo70 during axon formation.

## Arhgef7 acts upstream of Cdc42b

We have shown previously that active, fast-cycling TC10 F34L rescues the loss of Arhgef7 in neurons. In contrast, Cdc42u F28L was not able to restore axon formation although Arhgef7 is well established as a GEF for Cdc42 (Feng et al, 2006; Zhou et al, 2016; López Tobón et al, 2018). We first tested whether Arhgef7 could activate Cdc42b in a heterologous system. Cdc42b or Cdc42u was co-expressed with Arhgef7 in HEK293T cells and the level of active Cdc42 analyzed by a pull-down assay using the Cdc42- and Rac-interactive binding domain of PAK, which preferentially binds to GTP-bound Cdc42. Although Arhgef7 activated both Cdc42 isoforms, the increase in active Cdc42b (412.5%) was more pronounced than in Cdc42u (144.8%) (Fig 3A and B).

Because only Cdc42u was used for previous rescue experiments, we tested whether the Cdc42b isoform could rescue the loss of Arhgef7. The knockdown of Arhgef7 impaired axon formation in cultured hippocampal neurons from rat embryos as previously reported and could be rescued by an RNAi-resistant Arhgef7 (Fig S2C and D) (López Tobón et al, 2018). The efficiency of the knockdown construct was confirmed by Western blot after expression in HEK293T cells (Fig S2A and B). The knockdown of Arhgef7 reduced the number of neurons with a single axon from 77.9% ± 0.9% in controls to 50.9% ± 0.6%. Axon formation after the knockdown of Arhgef7 could be restored by the expression of TC10 F34L. The expression of TC10 F34L reduced the number of unpolarized neurons from 44.6% ± 0.6% after Arhgef7 knockdown to 21.3% ± 1%, similar to controls (18.8% ± 0.9% unpolarized neurons). Interestingly, Cdc42b F28L was also able to rescue the Arhgef7 knockdown (20.3% ± 1.4% unpolarized neurons) in contrast to Cdc42u F28L that was not able to restore axon formation (45.2% ± 0.8% unpolarized neurons; Fig 3C and D). These results show that Cdc42b acts downstream of Arhgef7 to promote axon formation in neurons.

---

sfGFP. **(G)** Hippocampal neurons from E18 rat embryos were co-transfected with control or knockdown vectors for Exo70 expressing H2B-RFP (yellow) and vectors for Exo70-sfGFP or truncated Exo70ΔNT (GFP: cyan). Neurons were analyzed at 3 d.i.v. with an anti-acetylated tubulin antibody as an axonal marker (magenta). Scale bar: 20 μm. **(H)** Percentage of neurons without an axon (0), with a single axon (1), and with multiple axons (>1) was quantified for (G) (an unpaired $t$ test, n = 3 independent experiments that are biological replicates, N = 312 neurons). Values are means ± SEM. ns, $P > 0.05$; *$P \leq 0.05$; **$P \leq 0.01$; ***$P \leq 0.001$; and ****$P \leq 0.0001$. Source data are available for this figure.

To address the role of Cdc42 palmitoylation, we mutated the cysteine residues C188 and C189 individually and in combination (Fig 2A). C189 is palmitoylated, and C188 can be modified by prenylation or palmitoylation (Wirth et al, 2013). Mutation of a single cysteine did not impair the ability to rescue the Arhgef7 knockdown, indicating that neither cysteine is essential for Cdc42b function by itself (Fig S2E and F). Mutation of C188 and C189 that both can be palmitoylated abolished the ability of Cdc42b to rescue the Arhgef7 knockout but also prevents membrane localization (Wirth et al, 2013). Taken together, these results indicate that membrane localization is essential but not the modification of a specific cysteine.

### Isoform-specific regulation of exocytosis

Because TC10 was shown to promote axon formation by stimulating the exocytosis of PPVs, we tested whether Cdc42b also regulates exocytosis in neurons. We generated knockdown constructs targeting TC10, Cdc42b, and Cdc42u and confirmed their efficacy and specificity by Western blots after expression in HEK293T cells (Fig S3A, B, and E–H). The knockdown constructs targeted sequences specific for the individual Cdc42 isoforms and did not suppress the other isoform (Fig S3I and J). As shown previously, the knockdown of TC10, Cdc4b, or Cdc4u impaired axon formation (Garvalov et al, 2007; Kang et al, 2008; Dupraz et al, 2009; Mukai et al, 2015; Yap et al, 2016). Axon formation was restored by the expression of their respective RNAi-resistant rescue constructs (Figs S3C and D and 4A and B). The combined knockdown of both Cdc42b and TC10 had a stronger effect than the single knockdowns, indicating that they may substitute for each other to some extent (Fig S3K and L).

Inhibition of palmitoylation blocks axon formation similar to the knockdown of TC10 and Cdc42 (Mukai et al, 2015). Therefore, we tested whether palmitoylation is required for the membrane recruitment of Exo70 (Fig 4C–E). Hippocampal neurons were treated with the palmitoylation inhibitor 2BP at 0 d.i.v., and the localization of Exo70 was analyzed at 1 d.i.v. by immunofluorescence with an antibody for Exo70. Inhibition of palmitoylation blocked the preferential localization of Exo70 to a single neurite and reduced the membrane-proximal localization of Exo70, indicating that palmitoylation is required for neuronal polarization and Exo70 function.

To analyze whether their loss affects the exocytosis of post-Golgi vesicles, hippocampal neurons were transfected with control or knockdown constructs targeting TC10, Cdc4b, or Cdc42u (Fig 4F and G). Neurons were labeled with BODIPY-ceramide at 1 d.i.v. and stimulated with Igf1. Exocytosis was quantified as a reduction in the signal for BODIPY-labeled vesicles in growth cones. In contrast to the Cdc42u knockdown, the Igf1-induced exocytosis was blocked to a large extent after the knockdown of either Cdc42b or TC10, where only a very small decrease in the signal for BODIPY-labeled vesicles was observed (Fig 4F and G). These results show that all three GTPases are involved in axon formation in developing neurons, but only TC10 and Cdc42b are required for the Igf1-induced exocytosis of post-Golgi vesicles. Although it is required for axon formation,

Cdc42u does not rescue the Arhgef7 knockdown and is also not essential for exocytosis.

Previous studies demonstrated that TC10, not Cdc42, recruits Exo70 to the membrane in mammals, whereas orthologs of Cdc42 perform this function in organisms such as *Saccharomyces cerevisiae* or *Drosophila* that do not contain a TC10 ortholog in their genome (Robinson et al, 1999; Zhang et al, 2001; Inoue et al, 2003; Boureux et al, 2007; Jones et al, 2014; Koon et al, 2018). However, only the Cdc42u isoform was tested in previous studies for its interaction with Exo70. Our results confirm these results for Cdc42u but show that the Cdc42b isoform can interact with Exo70, indicating that the isoforms perform different functions during axon formation. Cdc42b, like TC10, is required for axon formation and the Igf1-induced exocytosis of post-Golgi vesicles in the growth cones of differentiating neurons (Garvalov et al, 2007; Dupraz et al, 2009; Mukai et al, 2015; Yap et al, 2016). Taken together, our results show that Cdc42b promotes exocytosis and axon formation downstream of Arhgef7.

Cdc42b has been implicated previously in dendrite morphogenesis (Kang et al, 2008; Wirth et al, 2013; Yap et al, 2016; Lee et al, 2021). Interestingly, Arhgef7 and Exo70 also localize to dendritic spines, where they modulate spine density and synapse formation (Gerges et al, 2006; Sun & Bamji, 2011; Llano et al, 2015; Lira et al, 2018). The loss of Arhgef7, Cdc42b, or Exo70 at later stages of neuronal differentiation all leads to an impairment in spine morphogenesis (Kang et al, 2008; Wirth et al, 2013, 2022; Llano et al, 2015; Lira et al, 2018). This suggests that these proteins may have a similar function in regulating exocytosis also in mature dendrites.

TC10 serves as a regulator of exocytosis in different cell types (Chiang et al, 2001; Jiang et al, 2002; Inoue et al, 2003, 2006; Fujita et al, 2013). Our results show that Cdc42b performs a similar function in mammalian neurons. TC10 emerged together with Cdc42b in chordates, whereas only one Cdc42 isoform is present in eukaryotic lineages that lack TC10 orthologs (Boureux et al, 2007). The interaction of Cdc42b with Exo70 and its function in exocytosis indicate that the role of Cdc42 in the regulation of the exocyst is conserved also in mammals but restricted to the Cdc42b isoform.

The binding site of TC10 was mapped previously to the N-terminal half of Exo70 (Inoue et al, 2003). We could identify two domains in Exo70 within this sequence that interact with Cdc42b and TC10. Structural information is available for the yeast exocyst complex and yeast and mouse Exo70 but is missing for complexes with GTPases (Moore et al, 2007; Mei et al, 2018). These would help to understand the molecular basis of the differential interaction between the Cdc42 isoforms and Exo70. It will be interesting to determine whether the two GTPase binding sites in Exo70 can independently interact with two GTPase molecules or represent different interaction surfaces for a single GTPase. High-resolution structural studies are required to distinguish between these possibilities.

## Materials and Methods

### Reagents and tools table

| Reagent/Resource | Source | Identifier |
|---|---|---|
| **Experimental models** | | |
| HEK293T | MPI für Hirnforschung (Frankfurt) | N/A |
| Wistar rat | Harlan Winkelmann | N/A |
| Arhgef7$^{flox/flox}$; Emx1$^{Cre/+}$ (Arhgef7 cKO mouse) | López Tobón et al (2018) | N/A |
| **Recombinant DNA** | | |
| psfGFP-C1 | Pédelacq et al (2005) | Addgene (54579) |
| psfGFP-N1 | Pédelacq et al (2005) | Addgene (54737) |
| psfGFP-C3 | This study | N/A |
| pEGFP-C3-Exo70 | Martin et al (2014) | Addgene (53761) |
| pGEX6P1-GFP-Nanobody | Katoh et al (2015) | Addgene (61838) |
| pCMV-sfGFP-hsExo70 | This study | N/A |
| pCMV-hsExo70-sfGFP | This study | N/A |
| pCMV-hsExo70ΔNT-sfGFP | This study | N/A |
| psfGFP-Exo70-NT | This study | N/A |
| psfGFP-Exo70-NT + H1 | This study | N/A |
| psfGFP-Exo70-H1-H2 | This study | N/A |
| psfGFP-Exo70-H2-H6 | This study | N/A |
| psfGFP-Exo70-H3-H6 | This study | N/A |
| psfGFP-Exo70-H7-H10 | This study | N/A |
| psfGFP-Exo70-H11-H19 | This study | N/A |
| pMV-myc-TC10 | This study | N/A |
| pCMV-myc-TC10-Q67L | This study | N/A |
| pBK-eGFP-TC10-F34L | This study | N/A |
| pCMV-myc-Cdc42b | This study | N/A |
| pCMV-myc-Cdc42b-G12V | This study | N/A |
| pBK-eGFP-Cdc42b-F28L | This study | N/A |
| pCMV-myc-Cdc42u | This study | N/A |
| pCMV-myc-Cdc42u-G12V | This study | N/A |
| pBK-eGFP-Cdc42u-F28L | This study | N/A |
| HA-Arhgef7 | López Tobón et al (2018) | N/A |
| pcDNA6.2-GW/EmGFP-miR | Invitrogen | Invitrogen (K493600) |
| pcDNA6.2-H2B-mRFP-miR | di Meo et al (2021) | N/A |
| pcDNA6.2-EmGFP-miR-Exo70 | This study | N/A |
| pcDNA6.2-EmGFP-miR-Arhgef7 | This study | N/A |
| pcDNA6.2-EmGFP-miR-TC10 | This study | N/A |
| pcDNA6.2-EmGFP-miR-Cdc42b | This study | N/A |
| pcDNA6.2-EmGFP-miR-Cdc42u | This study | N/A |
| pcDNA6.2-H2B-mRFP-miR-Exo70 | This study | N/A |
| pcDNA6.2-H2B-mRFP-miR-Arhgef7 | This study | N/A |
| pcDNA6.2-H2B-mRFP-miR-TC10 | This study | N/A |
| pcDNA6.2-H2B-mRFP-miR-Cdc42b | This study | N/A |
| pcDNA6.2-H2B-mRFP-miR-Cdc42u | This study | N/A |
| pCMV6-mmExoc7-myc-DDK | OriGene | Cat# MR209780 |

| Reagent/Resource | Source | Identifier |
|---|---|---|
| **Experimental models** | | |
| pCMV-mmExo70-sfGFP | This study | N/A |
| psfGFP-Arhgef7 | This study | N/A |
| psfGFP-Arhgef7 RNAi-rescue | This study | N/A |
| psfGFP-TC10 | This study | N/A |
| psfGFP-TC10 RNAi-rescue | This study | N/A |
| pBK-eGFP-Cdc42b | This study | N/A |
| pBK-eGFP-Cdc42b-C188S-F28L | This study | N/A |
| pBK-eGFP-Cdc42b-C189S-F28L | This study | N/A |
| pBK-eGFP-Cdc42b-C188,189S-F28L | This study | N/A |
| pBK-eGFP-Cdc42b RNAi-rescue | This study | N/A |
| pBK-eGFP-Cdc42u | This study | N/A |
| pBK-eGFP-Cdc42u RNAi-rescue | This study | N/A |
| Antibodies | | |
| Mouse GFP monoclonal antibody (GF28R) | Thermo Fisher Scientific | Cat# A5-15256; RRID: AB_10979281 |
| Mouse anti-myc antibody (9E10) | Merck-Millipore | Cat# MABE282; RRID: AB_11213164 |
| Mouse anti-acetylated tubulin antibody | Sigma-Aldrich | Cat# T7451; RRID: AB_609894 |
| Rabbit anti–microtubule-associated protein 2 antibody | Merck Millipore | Cat# AB5622; RRID: AB_91939 |
| Rabbit Exo70 antibody | Proteintech | Cat# 12014-1-AP, RRID: AB_2101698 |
| Goat anti-mouse Alexa Fluor 350 (IgG2b) | Thermo Fisher Scientific | Cat# A-21140, RRID: AB_2535777 |
| Goat anti-rabbit Alexa Fluor 488 | Thermo Fisher Scientific | Cat# A-11008, RRID: AB_143165 |
| Goat anti-rabbit Alexa Fluor 647 | Thermo Fisher Scientific | Cat# A-21246, RRID: AB_2535814 |
| HRP anti-mouse | Jackson ImmunoResearch Labs | Cat# 115-035-003, RRID: AB_10015289 |
| **Oligonucleotides and other sequence-based reagents** | | |
| Oligonucleotides | This study | Table S1 |
| **Chemicals, enzymes, and other reagents** | | |
| Neurobasal medium (NBM) | Thermo Fisher Scientific | 21103049 |
| B-27 supplement | Thermo Fisher Scientific | 17504044 |
| Opti-MEM (reduced serum medium) | Thermo Fisher Scientific | 31985047 |
| Poly-L-ornithine hydrobromide | Sigma-Aldrich | P3655 |
| BODIPY FL C5-ceramide | Thermo Fisher Scientific | B22650 |
| Recombinant human Igf1 | Novus Biologicals | NBP2-35000 |
| 2-Bromohexadecanoic acid (2BP) | Sigma-Aldrich | 238422 |
| **Software** | | |
| ImageJ | NIH | RRID: SCR_002074; https://imagej.nih.gov/ij/ |
| Prism 7 | GraphPad Prism | Version 7.04 |
| ZEN 2.3 (blue edition) software | Zeiss | N/A |
| **Other** | | |
| UptiLight HRP blotting chemiluminescent substrate | Interchim | UP99619A |
| BLOCK-iT Pol II miR RNAi expression vector kit | Invitrogen | K493600 |
| Amaxa basic neuron SCN nucleofector kit | Lonza | VSPI-1003 |

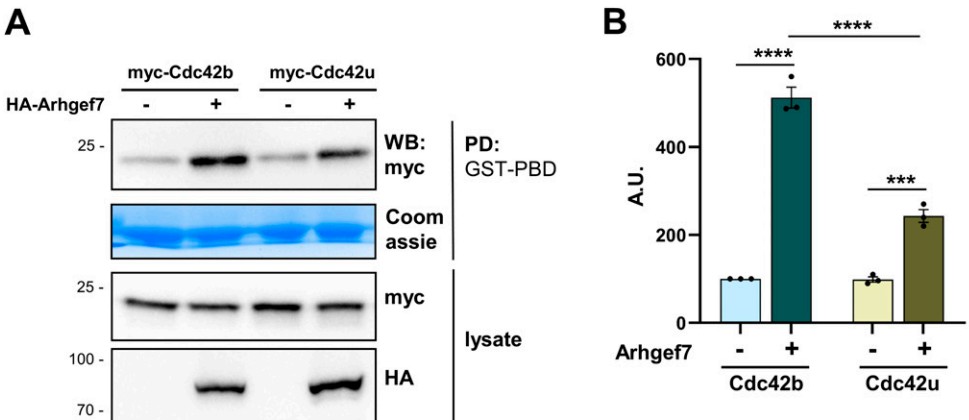

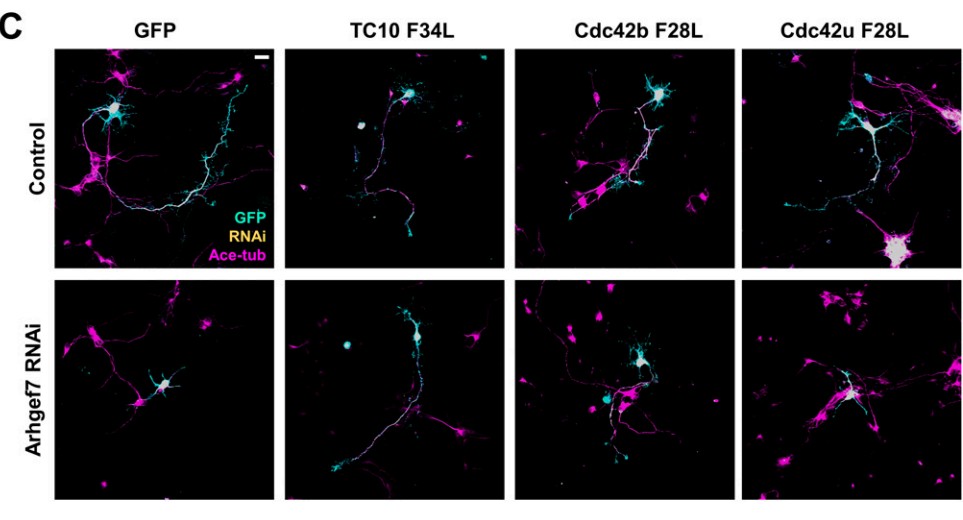

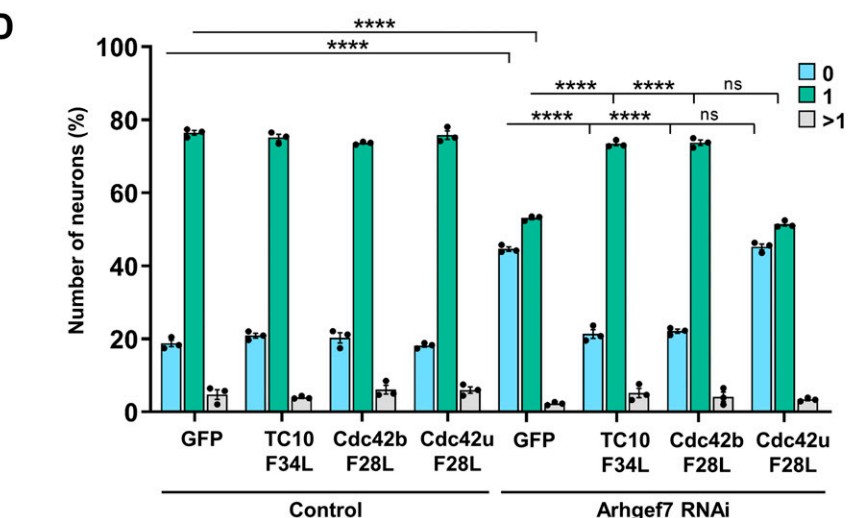

**Figure 3.    Active Cdc42b rescues the loss of Arhgef7 in neurons.**
**(A)** HA-Arhgef7 was co-expressed with myc-Cdc42b or myc-Cdc42u in HEK293T cells, and the amount of active Cdc42 was determined by a pull-down assay with GST-PAK-PBD (Coomassie) and Western blot (WB: anti-myc antibody). **(B)** The amount of active Cdc42 was quantified for (A) and normalized to total myc-Cdc42 (one-way ANOVA followed by Tukey's test, n = 3 independent experiments that are biological replicates). **(C)** Hippocampal neurons were co-transfected with control or knockdown vectors for Arhgef7 (H2B-RFP: yellow) and vectors for GTPases as indicated (GFP: cyan). Neurons were analyzed at 3 d.i.v. (acetylated tubulin: magenta). Scale bar: 20 μm. **(D)** Percentage of neurons without an axon (0), with a single axon (1), and with multiple axons (>1) was quantified for (C) (an unpaired t test, n = 3 independent experiments that are biological replicates, N = 274 neurons). Values are means ± SEM. ns, P > 0.05; *P ≤ 0.05; **P ≤ 0.01; ***P ≤ 0.001; and ****P ≤ 0.0001.
Source data are available for this figure.

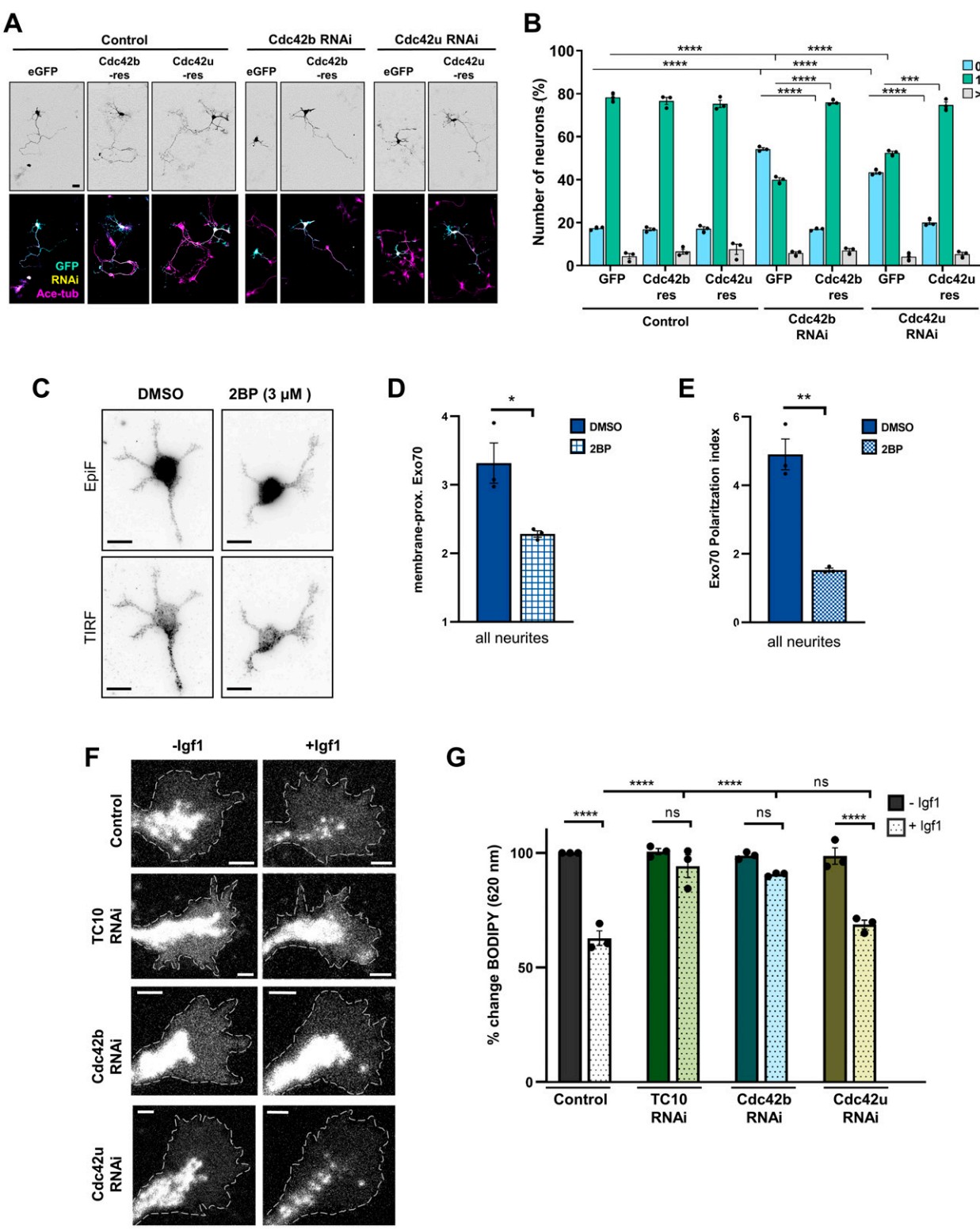

**Figure 4. Cdc42b is required for exocytosis and axon formation.**
**(A)** Cdc42b and Cdc42u are required for axon formation. Hippocampal neurons were co-transfected with control or knockdown vectors for Cdc42b or Cdc42u (H2B-RFP: yellow) and vectors for sfGFP or RNAi-resistant sfGFP-Cdc42b or sfGFP-Cdc42u (GFP: cyan). Neurons were analyzed at 3 d.i.v. (acetylated tubulin: magenta). Scale bar: 20 µm. **(B)** Percentage of neurons without an axon (0), with a single axon (1), and with multiple axons (>1) was quantified for (A) (an unpaired *t* test, n = 3 independent experiments that are biological replicates, N = 262 neurons). **(C)** Hippocampal neurons were treated with DMSO (vehicle) or 3 µM 2-bromopalmitate for 24 h and analyzed at 1 d.i.v. by staining with an anti-Exo70 antibody. Epifluorescence and TIRF signals indicate total and plasma membrane-proximal Exo70, respectively. Scale bar:

# Materials and Methods

## Experimental model and subject details

All animal protocols were performed in accordance with the guidelines of the North Rhine-Westphalia State Environment Agency (Landesamt für Natur, Umwelt und Verbraucherschutz). Both rats and mice were maintained at the animal facility of the Institute for Integrative Cell Biology and Physiology (University of Münster) under standard housing conditions with a 12-h light/dark cycle (lights on from 07:00 to 19:00 h) at a constant temperature (23°C) and with ad libitum supply of food and water. Timed pregnant rats were set up in-house. Pregnant rats were anesthetized by exposure to isoflurane followed by decapitation and primary cultures prepared from embryos at embryonic day 18 (E18). During dissection, neurons from all embryos (regardless of sex) were pooled. Generation and genotyping of the Arhgef7 conditional knockout mouse line has been described in a previous study (López Tobón et al, 2018).

## Neuronal cell culture and transfection

Primary cortical or hippocampal neurons from embryonic day 17 (E17) mice or E18 rat embryos were transfected by calcium phosphate co-precipitation. In brief, cortices were dissected, collected in HBSS (Invitrogen), and digested in trypsin (0.25% trypsin–EDTA; Invitrogen) for 8 min. Trypsin-digested tissue was mechanically dissociated in DMEM (1× DMEM [Invitrogen], 2 mM glutamine, 50 U/ml gentamycin, and 10% FCS) and passed through a cell strainer (40 μm pore size; Sarstedt). Dissociated neurons were seeded onto poly-L-ornithine–coated (10 μg/ml; Sigma-Aldrich) glass coverslips or eight-well chamber slides ($\mu$-Slide eight well; Ibidi) at a density of 35,000 cells/cm$^2$. Neurons were allowed to attach in the presence of DMEM for 1.5 h, and the medium was replaced by NBM (1× NBM [Invitrogen], 2 mM glutamine, 50 U/ml gentamycin, and 2% B27 [vol/vol]). For transfection, the culture medium was replaced 3 h after plating by Opti-MEM (Invitrogen) before the DNA mixture was added. After incubation for 45 min at 37°C at 5% $CO_2$, neurons were washed for 15 min with 1 ml of Opti-MEM, which had been preincubated at 37°C at 10% $CO_2$, and the conditioned NBM was added back to the cells. Depending on the experiment, hippocampal and cortical neurons were transfected at 0 d.i.v. and fixed at either 1 or 3 d.i.v.

For analysis of exocytosis at 1 d.i.v., primary hippocampal neurons were transfected by nucleofection using Nucleofector II (Amaxa) and the Amaxa Basic Neuron SCN Nucleofector kit (Lonza). For one 35-mm dish ($\mu$-Dish Quad; Ibidi), $2 \times 10^5$ cells were resuspended in 20 $\mu$l of Basic Neuron SCN Nucleofector solution, with SCN supplement and nucleofection was performed using the SCN Basic Neuron Program 1, with 0.6 $\mu$g of plasmid DNA.

## Plasmids

pEGFP-C3-Exo70 was a gift from Channing Der (plasmid #53761; Addgene). psfGFP-C1 (plasmid #54579; Addgene) and psfGFP-N1 (plasmid #54737; Addgene) were obtained from Michael Davidson and Geoffrey Waldo, and pGEX6P1-GFP-Nanobody (plasmid #61838; Addgene) was obtained from Kazuhisa Nakayama. psfGFP-C2 was generated by modifying psfGFP-C1, using primers described in Table S1 that introduced a different reading frame. Constructs expressing human and mouse Exo70 were amplified by PCR from pEGFP-C3-Exo70 and pCMV6-mmExoc7-myc-DDK (Cat# MR209780; OriGene), respectively, and inserted into psfGFP-C3 (pCMV-sfGFP-hsExo70) or psfGFP-N1 (pCMV-hsExo70-sfGFP and pCMV-mmExo70-sfGFP). The constructs expressing the different fragments of Exo70-(NT (aa 1–84), NT + H1 (aa 1–106), H1-H2 (aa 85–106), H2-H6 (aa 107–240), H3-H6 (aa 133–240), H7-H10 (aa 241–404), and H11-H19 (aa 405–653)) and truncated Exo70ΔNT were amplified by PCR using primers described in Table S1 and cloned into psfGFP-C3. The expression vectors for pGST-PAK-PBD, Cdc42u-G12V, Cdc42u-F28L, TC10-Q67L, TC10 F34L, and Arhgef7 have been described before (Bagrodia et al, 1995, 1998; Lin et al, 1999; López Tobón et al, 2018). The expression vectors for Cdc42b, Cdc42b-G12V, and Cdc42b-F28L were amplified by PCR from the corresponding Cdc42u expression vectors using primers described in Table S1.

Loss-of-function experiments were performed with miRNAs generated using the BLOCK-iT Pol II miRNA Expression Vector Kit (Invitrogen) and oligonucleotides that are mentioned in Table S1. Target sequences were cloned into the pcDNA6.2-GW/EmGFP-miR expression plasmid following the manufacturer's instructions. Individual H2B- and RFP-tagged RNAi constructs were generated as previously described (di Meo et al, 2021). RNAi-resistant expression vectors for Arhgef7 and TC10 were generated by site-directed mutagenesis using Phusion High-Fidelity DNA Polymerase (Thermo Fisher Scientific) and mutagenesis primers described in Table S1 to introduce two mutations on the binding sites of the respective miRNAs of these genes. For Cdc42b and Cdc42u, RNAi-resistant expression vectors were generated by PCR amplification, where mutations were introduced in the reverse primers (Table S1) to impair miRNA binding to the target sequences.

## Immunofluorescence staining

Immunofluorescence staining of cultured neurons was carried out as described (di Meo et al, 2021). Primary cortical or hippocampal neurons were fixed with 4% paraformaldehyde and 4% sucrose for 15 min at 37°C, quenched in 50 mM ammonium chloride at RT for

---

10 μm. **(D)** Intensity of the total internal reflection fluorescence microscopy signal in neurites was normalized to epifluorescence signals (total Exo70) and area. The average relative intensity for membrane-proximal Exo70 in all neurites is shown. **(E)** Polarization index for membrane-proximal Exo70 was calculated as the normalized TIRF intensity of the brightest neurite relative to the average of the normalized TIRF intensities of the other neurites (unpaired $t$ test, N = 24 neurons from n = 3 independent experiments). **(F)** Hippocampal neurons were co-transfected with control or knockdown vectors for TC10, Cdc42b, or Cdc42u. Neurons were incubated with BODIPY FL C5-ceramide at 1 d.i.v. to label post-Golgi vesicles and cultured in medium without (−Igf1) or with (+Igf1) 20 nM Igf1 for 10 min. A decrease in the fluorescence intensity at 620 nm indicates the exocytosis of post-Golgi vesicles. Scale bar: 2 μm. **(G)** Igf1-stimulated exocytosis was quantified for (F) as the relative change in the fluorescence intensity for BODIPY-labeled vesicles in growth cones at 620 nm normalized to the signal at 515 nm. The value for control neurons (−Igf1) was set as 100% (one-way ANOVA followed by Tukey's test, N = 24 neurons from each condition from n = 3 independent experiments that are biological replicates). Values are means ± SEM. ns, $P > 0.05$; *$P \leq 0.05$; **$P \leq 0.01$; ***$P \leq 0.001$; and ****$P \leq 0.0001$.
Source data are available for this figure.

10 min, permeabilized with 0.1% Triton X-100 in 1× PBS for 3 min, and treated with blocking solution (2% normal goat serum, 2% bovine serum albumin, and 0.2% fish gelatin in PBS) for 1 h at RT. Primary antibody incubation was performed overnight at 4°C. Cells were washed with 1× PBS and incubated with the appropriate Alexa Fluor–conjugated secondary antibody (Thermo Fisher Scientific) at RT for 30 min, and mounted using Mowiol (Sigma-Aldrich). Antibodies were diluted in 10% blocking solution.

### BODIPY-ceramide labeling, and image acquisition and analysis

Primary cortical or hippocampal neurons were labeled with BODIPY FL $C_5$-ceramide at 1 d.i.v. as reported (Wang et al, 2011). Briefly, neurons were incubated with 5 μM BODIPY FL $C_5$-ceramide (Invitrogen) for 30 min at 37°C, followed by two washes with conditioned NBM and subsequent incubation at 37°C for 1 h. After stimulation of PPV exocytosis for 10 min with 20 nM Igf1 (Invitrogen) at 37°C, neurons were fixed and imaged with a Plan-Apochromat 40×/1.3 Oil DIC M27 objective and a Zeiss LSM 800 confocal laser scanning microscope. Fixed neurons were imaged using the 488-nm laser line and fluorescence signals collected with individual filter sets at 515 and 620 nm. The fluorescence intensities of individual channels were quantified in growth cones of these neurons using ImageJ (NIH). The signals at 620 nm were normalized to those at 515 nm.

### Transfection of HEK293T cells and Western blot

HEK293T cells were transfected by calcium phosphate co-precipitation. Transfected HEK293T cells were harvested in ice-cold PBS and lysed at 4°C for 1 h (lysis buffer: Tris–HCl 50 mM, pH 7.4, NaCl 150 mM, DTT 1 mM, MgCl$_2$ 1.5 mM, EDTA 4 mM, glycerol 10% [vol/vol], Triton X-100 1% [vol/vol], complete protease inhibitor [Merck], and tyrosine and serine/threonine phosphatase inhibitors [Sigma-Aldrich]). After addition of 5× Laemmli buffer and boiling at 95°C, proteins were separated by SDS–PAGE and transferred to PVDF membranes. Non-specific binding was blocked at RT with 5% non-fat dry milk in Tris-buffered saline and 0.1% Tween-20 (TBS-T) for 1 h. The membrane was incubated overnight with primary antibodies diluted in blocking solution at 4°C. After several washes with TBS-T, membranes were incubated with HRP-coupled secondary antibodies (Jackson ImmunoResearch Labs) for 2 h at RT. Peroxidase activity was visualized by the enhanced chemiluminescence detection system (Interchim) using the ChemiDocTM MP imaging system (Bio-Rad).

### Biochemistry

For co-immunoprecipitation experiments, a GST-tagged anti-GFP nanobody was expressed in *Escherichia coli* BL21 cells as described (Katoh et al, 2015). The anti-GFP nanobody was coupled to glutathione Sepharose 4B beads (VWR) and incubated with lysates from transfected HEK293T cells for 1 h at 4°C. The beads were washed thrice (10 min each with Tris–HCl, pH 7.4, 50 mM, NaCl 250 mM, DTT 1 mM, MgCl$_2$ 1.5 mM, EDTA 4 mM, glycerol 10% [vol/vol], Triton X-100 0.1% [vol/vol], and complete protease inhibitor), and bound proteins were eluted using 2× Laemmli buffer and analyzed by Western blot. To determine the amount of active Cdc42, a pull-down assay was performed with the GTPase binding domain (PBD) from PAK3 (GST-PAK-PBD). GST-PAK-PBD that was expressed in *E. coli* BL21 cells was coupled to glutathione Sepharose beads (VWR). The beads were incubated with lysates of transfected HEK293T cells, and the bound proteins were eluted with 2× Laemmli buffer and analyzed by Western blot.

### Image acquisition and analysis

Images were acquired on a Zeiss LSM 700 or LSM 800 confocal laser scanning microscope and processed and analyzed using ImageJ (NIH). For TIRF imaging of neurons at 1 d.i.v., the Olympus TIRF 4Line microscope was used with the Olympus UApoN 100×/1.49 NA oil objective.

### Quantification

Neuronal polarity was analyzed at 3 d.i.v. as described previously (Dhumale et al, 2018). Neurons were categorized as neurons showing only acetylated-tubulin–negative neurites (without an axon (0)), and neurons with a single (1) or with multiple (>1) acetylated-tubulin–positive axons. At least 200 cells per condition were counted for each experiment. The polarization index for membrane-proximal Exo70 was calculated as the TIRF intensity of the brightest neurite (normalized to the epifluorescence signal and area) relative to the average of the normalized TIRF intensities of the other neurites as described before (Grassi et al, 2015).

### Statistical analysis

All plotted values are expressed as means ± SEM. All experiments were performed independently at least three times. Specific numbers can be found in the figure legends. If not otherwise indicated, comparisons between two groups were performed using an unpaired *t* test, and comparisons of more than two groups were done using one-way ANOVA followed by Tukey's test with GraphPad Prism. No statistical methods were used to predetermine the sample size. The data were assumed to have a normal distribution, but this was not formally tested. Blinding was not performed because the genotype had to be determined before the experiment. For all analyses performed, significance was defined as ns, $P > 0.05$; $*P \leq 0.05$; $**P \leq 0.01$; $***P \leq 0.001$; and $****P \leq 0.0001$. When not indicated, the conditions did not show a significant difference.

## Data Availability

Data sets generated from the current study are available from the corresponding author upon reasonable request.

## Supplementary Information

# Acknowledgements

We thank Maria Wenning, Ina Kowsky, and Verena Stegemann for technical assistance. This work was supported by the Deutsche Forschungsgemeinschaft (DFG) through the Cells-in-Motion Cluster of Excellence (EXC 1003-CiM) and SFB 1348. The graphical abstract was created with BioRender.com (P.R. license agreement SO24QK598K).

## Author Contributions

P Ravindran: conceptualization, data curation, formal analysis, investigation, methodology, and writing—original draft, review, and editing.

AW Püschel: conceptualization, data curation, formal analysis, supervision, funding acquisition, validation, visualization, project administration, and writing—review and editing.

## Conflict of Interest Statement

The authors declare that they have no conflict of interest.

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
