## [Reviewer comments · Life Science Alliance]

An isoform-specific function of Cdc42 in regulating mammalian Exo70 during axon formation

Priyadarshini Ravindran and Andreas Puschel
DOI: <https://doi.org/10.26508/lsa.202201722>

Corresponding author(s): Andreas Puschel, University of Münster

Review Timeline:	Submission Date:	2022-09-13
	Editorial Decision:	2022-09-13
	Revision Received:	2022-12-07
	Editorial Decision:	2022-12-08
	Revision Received:	2022-12-09
	Accepted:	2022-12-09

Transaction Report:

Please note that the manuscript was previously reviewed at another journal and the reports were taken into account in the decision-making process at Life Science Alliance.

Referee #1 Review

Report for Author:

During neuronal polarization, one neurite elongates more than the other and becomes the axon. One important question in neurobiology is the mechanism by which only one neurite elongates during this transitional period. In this study, Priyadarshini Ravindran & Andreas W. Püschel show that an isoform of Cdc42 (Cdc42b) regulates axon formation. Specifically, Cdc42b mediates exocytosis throughout the interaction of Exo70 and downstream of Arhgef7.

First, it is highly commendable that an effort was made to tackle such a fundamentally important problem in neuronal polarity. The authors used a robust approach to test their hypothesis, supplemented with a blend of primary culture experiments that are well executed and for the most part, the data is nicely presented. Having said this, I consider that this manuscript does not reach the novelty required to be published in this journal given that this journal states on its website: "This journal publishes both long- and short-format papers that communicate major findings, offering novel physiological/functional insight of wide interest that is robustly documented by independent lines of evidence". Basically, we just learned with this manuscript that the effect of Cdc42 in axon elongation is isoform-specific.

Minor points to be considered are listed below:

1. The summary is mostly describing not the results from this manuscript. Only the last five lines described the actual results from this study
2. Some paragraphs in the results section should be considered as discussion (e.g., the first paragraph of page 6)

Referee #2 Review

Report for Author:

Ravindran and Püschel reveal that the brain-specific isoform of Cdc42, Cdc42b, is efficiently activated by Arhgef7 and contributes to axon formation via binding to Exo70, similar to TC10. In contrast, the ubiquitously expressed splice variant of Cdc42, Cdc42u, is not able to efficiently work in this pathway. The experiments presented are meaningful, well executed, and

properly interpreted. The results are interesting, but what needs to be elucidated more is the underlying mechanism.

Major comments:

1. Why do Cdc42u and Cdc42b interact differentially with ArhGEF7 and Exo70 although the splice variants are structurally extremely similar? A likely candidate could be the distinguishing palmitoylation and correspondingly a different membrane localization. It would be therefore interesting to investigate the presence of Cdc42u, Cdc42b, TC10, and Arhgef7 in lipid rafts in the presence and absence of IGF1. Will palmitoylation inhibitors abrogate the function of Cdc42b and TC10?
2. It would be also interesting to see whether inhibition of Cdc42b and TC10 have additive effects.

Referee #3 Review

Report for Author:

Ravindram and Puschel present clear evidence that the function of Cdc42 in regulating exocytosis is conserved in mammals and is specific of Cdc42b. The study is mostly based in biochemical analyses that are well designed and controlled. Each experiment provides strong evidence for the conclusions that are drawn. Although the study is important to the specific field of axon formation/growth, it lacks a broad biological significance and the novelty is somehow limited. In this respect, it would be better suited for a more specialized audience.

September 13, 2022

Re: Life Science Alliance manuscript #LSA-2022-01722-T

Andreas W. Puschel
Universität Münster
Molekularbiologie
Institut für Allg. Zoologie & Genetik
Schlossplatz 5
Münster 48149
Germany

Dear Dr. Puschel,

Thank you for submitting your manuscript entitled "An isoform-specific function of Cdc42 in regulating mammalian Exo70 during axon formation" to Life Science Alliance. We invite you to submit a revised manuscript addressing the Reviewer comments.

Thank you for this interesting contribution to Life Science Alliance. We are looking forward to receiving your revised manuscript.

Sincerely,

Eric Sawey, PhD
Executive Editor
Life Science Alliance
<http://www.lsa-journal.org>

B. MANUSCRIPT ORGANIZATION AND FORMATTING:

We thank the reviewers for their helpful comments and have modified the manuscript according to their suggestions.

Referee #1:

First, it is highly commendable that an effort was made to tackle such a fundamentally important problem in neuronal polarity. The authors used a robust approach to test their hypothesis, supplemented with a blend of primary culture experiments that are well executed and for the most part, the data is nicely presented.

Basically, we just learned with this manuscript that the effect of Cdc42 in axon elongation is isoform-specific.

We thank the reviewer for the positive comments but respectfully disagree with the notion that our manuscript only shows "that the effect of Cdc42 in axon elongation is isoform-specific". While the highly conserved Cdc42 regulates cell polarity in yeast and *Drosophila* through the exocyst complex, it was thought that - surprisingly - this is not the case in mammals. The vertebrate-specific Cdc42 isoforms have been known for a long time but it remained unclear how they perform different functions. Our manuscript identifies a molecular mechanism that explains how Cdc42b mediates its previously unrecognized function in regulating exocytosis. The results show that the interaction with the exocyst is conserved in mammals but specific to the Cdc42b isoform.

Minor points to be considered are listed below:

1. The summary is mostly describing not the results from this manuscript. Only the last five lines described the actual results from this study

1) We modified the abstract as suggested

2. Some paragraphs in the results section should be considered as discussion (e.g., the first paragraph of page 6).

2) The manuscript was modified as suggested and the paragraph was moved to the discussion at the end of the manuscript.

Referee #2:

Ravindran and Püschel reveal that the brain-specific isoform of Cdc42, Cdc42b, is efficiently activated by Arhgef7 and contributes to axon formation via binding to Exo70, similar to TC10. In contrast, the ubiquitously expressed splice variant of Cdc42, Cdc42u, is not able to efficiently work in this pathway. The experiments presented are meaningful, well executed, and properly interpreted. The result are interesting, but what needs to be elucidated more is the underlying mechanism.

Major comments:

1. Why do Cdc42u and Cdc42b interact differentially with ArhGEF7 and Exo70 although the splice variants are structurally extremely similar? A likely candidate could be the distinguishing palmitoylation and correspondingly a different membrane localization. It would be therefore interesting to investigate the presence of Cdc42u, Cdc42b, TC10, and Arhgef7 in lipid rafts in the presence and absence of IGF1. Will palmitoylation inhibitors abrogate the function of Cdc42b and TC10?

1) The reviewer suggested that the differential interaction of Cdc42 isoforms with Exo70 could depend on palmitoylation and mediate the localization to lipid rafts. Our results (Fig. 3A, B) show that both Cdc42 isoforms can be activated by Arhgef7, indicating that palmitoylation is not required for their interaction with Arhgef7. A previous study already demonstrated that a palmitoylation inhibitor blocks axon formation similar to the knockdown of TC10 and Cdc42 (Mukai et al., 2015). Therefore, we tested if palmitoylation is required for the membrane recruitment of Exo70 to establish a link between palmitoylation, axon formation and Exo70. The results (new Fig 4C-E; paragraph 2 on p. 8) show that the treatment of neurons with the palmitoylation inhibitor 2-BP blocks the preferential localization of Exo70 to a single neurite and reduces its membrane-proximal localization of Exo70 indicating that palmitoylation is required for neuronal polarization and Exo70 function. It has also been shown that palmitoylation directs GTPases including TC10 and Cdc42b to specific membrane domains often called lipid rafts (Watson et al., 2001, 2003; Wirth et al., 2013). The localization to lipid rafts is assayed biochemically by determining the distribution of membrane proteins to detergent resistant membrane fractions (DRMs) after extraction, sucrose density gradient centrifugation and Western blot (Klotzsch and Schütz, 2013). The presence of palmitoylated GTPases in DRMs has been demonstrated before but this result does not provide evidence for their localization to specific lipid domains in native membranes in living cells (Klotzsch and Schütz, 2013; Levental et al., 2020; Sezgin et al. 2017). The physiological relevance of DRM localization is also discussed controversially. The pathway that promotes axon formation acts in the nascent axon and differences in the localization to specific membrane domains will be restricted to the growth cone (Dupraz et al., 2009). Thus, any changes in the distribution to the DRMs will be limited to a small fraction of Cdc42 and difficult to detect. The isoform-specific antibodies also do not work well enough in Western blot to reliably detect endogenous levels of Cdc42 in neurons.

To address the role of Cdc42 palmitoylation we, therefore, mutated the cysteine residues C188 and C189 at the C-terminus individually and in combination. C189 is palmitoylated and C188 is modified by prenylation or palmitoylation (Wirth et al., 2013). The function of the mutants was analyzed by their ability to restore axon formation after knockdown of Arhgef7 (new Fig S2E and F; last paragraph on p. 7). Mutation of a single cysteine did not impair the ability to rescue the Arhgef7 knockdown. This rescue indicates that neither cysteine is essential for Cdc42b function but does not exclude the possibility that palmitoylation is required for its function since both residues can be palmitoylated (Wirth et al., 2013). Mutation of both cysteines abolished the ability of Cdc42b to rescue the Arhgef7 knockout but also prevents membrane localization (Wirth et al., 2013). Taken together, these results indicate that membrane localization is essential but not the modification of a specific cysteine residue. An ultrastructural analysis of Exo70/Cdc42 complexes is required to elucidate the molecular details of the interaction but this would go beyond the scope of this study.

2. It would be also interesting to see whether inhibition of Cdc42b and TC10 have additive effects.

2) We analyzed the combined knockdown of both Cdc42b and TC10 as suggested. Our results (new Fig S3K and L; first paragraph on p. 8) show that the combined knockdown has a stronger effect than the single knockdowns, indicating that they may substitute for each other to some extent.

References

- Klotzsch E, Schütz GJ (2012). A critical survey of methods to detect plasma membrane rafts. *Philos Trans R Soc Lond B Biol Sci* 368, 20120033. doi: 10.1098/rstb.2012.0033
- Levental I, Levental KR, Heberle FA (2020). Lipid Rafts: Controversies Resolved, Mysteries Remain. *Trends Cell Biol* 30, 341-353.
- Sezgin E, Levental I, Mayor S, Eggeling C (2017). The mystery of membrane organization: composition, regulation and roles of lipid rafts. *Nat Rev Mol Cell Biol* 18, 361-374.
- Watson RT, Shigematsu S, Chiang SH, Mora S, Kanzaki M, Macara IG, Saltiel AR, Pessin JE (2001). Lipid raft microdomain compartmentalization of TC10 is required for insulin signaling and GLUT4 translocation. *J Cell Biol* 154, 829-840.
- Watson RT, Furukawa M, Chiang SH, Boeglin D, Kanzaki M, Saltiel AR, Pessin JE (2003). The exocytotic trafficking of TC10 occurs through both classical and nonclassical secretory transport pathways in 3T3L1 adipocytes. *Mol Cell Biol* 23, 961-974.

Referee #3:

Ravindram and Puschel present clear evidence that the function of Cdc42 in regulating exocytosis is conserved in mammals and is specific of Cdc42b. The study is mostly based in biochemical analyses that are well designed and controlled. Each experiment provides strong evidence for the conclusions that are drawn. Although the study is important to the specific field of axon formation/growth, it lacks a broad biological significance and the novelty is somehow limited. In this respect, it would be better suited for a more specialized audience.

We think that our results are relevant not only for the question how axon formation is regulated. The function of Cdc42b is not restricted to axon formation and it has also been implicated in regulating activity-dependent plasticity of dendritic spines and synapses (Kang et al., 2008; Moutin et al., 2016; Wirth et al., 2022). It was thought that the highly conserved Cdc42 regulates cell polarity in yeast and *Drosophila*, but not in mammals, through the exocyst complex. The vertebrate-specific Cdc42 isoforms have been known for a long time but their relevance remained unclear. Our manuscript identifies a molecular mechanism that explains how Cdc42b mediates its previously unrecognized function in regulating exocytosis through the exocyst. This regulation Cdc42 is conserved in mammals but specific to the Cdc42b isoform.

December 8, 2022

RE: Life Science Alliance Manuscript #LSA-2022-01722-TR

Prof. Andreas W. Puschel
University of Münster
Institut für Integrative Zellbiologie und Physiologie
Schlossplatz 5
Münster 48149
Germany

Dear Dr. Puschel,

Thank you for submitting your revised manuscript entitled "An isoform-specific function of Cdc42 in regulating mammalian Exo70 during axon formation". We would be happy to publish your paper in Life Science Alliance pending final revisions necessary to meet our formatting guidelines.

- please add ORCID ID for corresponding author; you should have received instructions on how to do so
- please add the Twitter handle of your host institute/organization as well as your own or/and one of the authors in our system

A. FINAL FILES:

B. MANUSCRIPT ORGANIZATION AND FORMATTING:

****It is Life Science Alliance policy that if requested, original data images must be made available to the editors. Failure to provide**

original images upon request will result in unavoidable delays in publication. Please ensure that you have access to all original data images prior to final submission.**

The license to publish form must be signed before your manuscript can be sent to production. A link to the electronic license to publish form will be sent to the corresponding author only. Please take a moment to check your funder requirements.

Sincerely,

December 9, 2022

RE: Life Science Alliance Manuscript #LSA-2022-01722-TRR

Prof. Andreas W. Puschel
University of Münster
Institut für Integrative Zellbiologie und Physiologie
Schlossplatz 5
Münster 48149
Germany

Dear Dr. Puschel,

Thank you for submitting your Research Article entitled "An isoform-specific function of Cdc42 in regulating mammalian Exo70 during axon formation". It is a pleasure to let you know that your manuscript is now accepted for publication in Life Science Alliance. Congratulations on this interesting work.

DISTRIBUTION OF MATERIALS:

Again, congratulations on a very nice paper. I hope you found the review process to be constructive and are pleased with how the manuscript was handled editorially. We look forward to future exciting submissions from your lab.

Sincerely,
